# The Risk of *Clostridioides difficile* Recurrence after Initial Treatment with Vancomycin or Fidaxomicin Utilizing Cerner Health Facts

**DOI:** 10.3390/antibiotics11030295

**Published:** 2022-02-23

**Authors:** Ronald G. Hall, Travis J. Cole, Chip Shaw, Carlos A. Alvarez

**Affiliations:** 1Department of Pharmacy Practice, Jerry H. Hodge School of Pharmacy, Texas Tech University Health Sciences Center, Dallas, TX 75235, USA; carlos.alvarez@ttuhsc.edu; 2Clinical Research Data Warehouse, Texas Tech University Health Sciences Center, Lubbock, TX 79430, USA; travis.j.cole@ttuhsc.edu (T.J.C.); chip.shaw@ttuhsc.edu (C.S.)

**Keywords:** fidaxomicin, vancomycin, *Clostridioides difficile*, CDI, recurrence

## Abstract

(1) Background: Fidaxomicin has been shown to significantly reduce *Clostridioides difficile* infection (CDI) recurrences rates in randomized, controlled trials. However, national data from the Veterans Affairs has called the real-world applicability of these findings into question. Therefore, we conducted a retrospective cohort study of patients receiving fidaxomicin or vancomycin as initial therapy for an index case of CDI in the hospital to evaluate the relative rates CDI recurrence within 90 days of an index case. (2) Methods: We retrieved patients 18 years and older who were admitted between July 2011 through June 2018 and diagnosed and treated for CDI with vancomycin or fidaxomicin. The first occurrence of CDI with treatment was designated as the index case. Patients with CDI within 1 year prior to index case were excluded. From the remaining index cases (vancomycin = 14,785; fidaxomicin = 889) the primary outcome (a recurrence of CDI within 90 days of the index case) was determined. The CDI recurrence rates for fidaxomicin and vancomyicn were evaluated using a Cox Proportional Hazards model on a propensity score matched cohort. (3) Results: A statistically significantly lower risk of CDI recurrence was observed with fidaxomicin use in the matched cohort (889 patients per treatment) using a Cox Proportional Hazards model (HR 0.67, 95% CI 0.50–0.90). (4) Conclusions: Fidaxomicin was independently associated with a decreased CDI recurrence, as defined by readmission for CDI within 90 days.

## 1. Introduction

Fidaxomicin and vancomycin are recommended by multiple *Clostridioides difficile* infection (CDI) guidelines including the American College of Gastroenterology, European Society of Clinical Microbiology and Infectious Diseases, and Infectious Diseases Society of America/Society for Healthcare Epidemiology of America as first-line therapies for the initial episode of CDI [1,2,3]. However, the conclusions reached about fidaxomicin’s relative efficacy and/or role as the preferred CDI agent by the guidelines panels from evaluating the available data are not the same. Meta-analyses of four randomized controlled trials conducted by the Infectious Diseases Society of America/Society for Healthcare Epidemiology of America guideline panel suggest that fidaxomicin produces superior sustained clinical response (initial clinical response without subsequent recurrent symptoms) rates at four weeks after the end of treatment (Relative Risk [RR] 1.16, 95% Confidence Interval [CI] 1.09–1.24) [1,4,5,6]. The European Society of Clinical Microbiology and Infectious Diseases excluded an open-label trial of extended-pulsed fidaxomicin used by IDSA and found that fidaxomicin had similar treatment responses to vancomycin (RR 1.03, 95% CI 0.98–1.09) while significantly reducing recurrence at 28 days (RR 0.61, 95% CI 0.47–0.81) [3,5]. The American College of Gastroenterology guidelines highlight similar outcomes at 90 days for fidaxomicin and vancomycin in terms of recurrence (24.4% for both) and death (23 vs. 22%, *p* = 0.85) in a propensity-matched national cohort from the Veterans Affairs (VA) [2,3].

The findings from the national VA cohort create uncertainty regarding whether the improvements in recurrence observed in randomized, controlled trials can be replicated in standard medical practice outside of a randomized, controlled trial. A national multi-center evaluation of fidaxomicin’s impact on CDI recurrence outside of the VA system has not been conducted to our knowledge.

Therefore, we conducted a retrospective cohort study of patients receiving fidaxomicin or vancomycin as initial therapy for an index case of CDI in the hospital. This study evaluated the impact of CDI treatment choice (fidaxomicin vs. vancomycin) on recurrence rates in hospitalized patients using Cerner Health Facts^®^.

## 2. Results

### 2.1. Baseline Characteristics

Of the 15,674 eligible patients, 889 (5.7%) were given fidaxomicin for CDI and 14,785 (94.4%) were given oral vancomycin. The mean age of patients was 66.7 years, and 42.0% were male. We matched 889 patients who received fidaxomicin to 889 patients who received vancomycin. Table 1 shows the standardized differences between key variables both before and after propensity matching. The standardized differences became smaller in the matched cohort for almost all variables, as expected. There were no statistically significant differences between groups for any of the key characteristics after matching.

Table 1 also demonstrates the breakdown by treatment group for both the entire cohort and for the matched patients. Although there were several statistically significant differences between receipt versus nonreceipt of fidaxomicin, there were no clinically significant differences >3% between groups. Unadjusted 90-day recurrence was 8.3% for those who received fidaxomicin versus 12.3% in those who received vancomycin (*p* < 0.001).

### 2.2. Univariable Results

Table 2 contains the univariable results regarding the risk of CDI recurrence. Overall, 12% of patients experienced a CDI recurrence. Several factors were statistically significantly associated with CDI recurrence in the univariable analysis in the entire cohort including male sex, race, census region, rural treatment facility, van Walraven score, severe CDI (33% vs. 27%, *p* < 0.001), PPI use during CDI, and prior antibiotic use. Fulminant CDI (0.4% vs. 0.6%, *p* = 0.44) and H2 antagonist use (14% vs. 15%, *p* = 0.30) were not associated with CDI recurrence in the entire cohort.

### 2.3. Primary Outcome

A statistically significantly lower risk of CDI recurrence was observed with fidaxomicin use in the matched cohort using a Cox Proportional Hazards model (HR 0.67, 95% CI 0.50–0.90). The adjusted Kaplan–Meier graph depicting these findings is shown in Figure 1.

## 3. Discussion

This study was designed to determine whether fidaxomicin is better than vancomycin with regard to 90-day recurrence, as measured by hospital readmission with an ICD-9 or ICD-10 code for CDI. Fidaxomicin did produce statistically significantly lower recurrence rates compared to vancomycin in our national cohort of facilities participating in Cerner HealthFacts^®^. To our knowledge, this is the largest evaluation of fidaxomicin for CDI recurrence in the literature.

Our finding that fidaxomicin significantly decreases CDI recurrence rates compared with vancomycin agrees with the European Society of Clinical Microbiology and Infectious Diseases meta-analysis as well as the individual studies used by the meta-analysis [1,3,4,5,6,7]. This finding is in contrast to fidaxomicin having no impact on recurrence in the VA cohort [8]. This may partially be because our cohort included non-severe CDI whereas the VA study only evaluated severe CDI cases. All of these studies observed higher recurrence rates than our cohort. This is likely because our definition of recurrence required rehospitalization with an ICD-9 or ICD-10 code for CDI within 90 days of the end of inpatient therapy for the index CDI episode. A previous single-center study also demonstrated a significant reduction in 90-day CDI readmissions [9]. Differences between this study and ours include that 63% of the patients in their study were treated for a recurrent CDI episode and their cohort included a higher percentage of patients with moderate or severe CDI (60%). A study of fidaxomicin and vancomycin in two New York VA hospitals also showed that fidaxomicin reduced 60-day recurrence (8 vs. 22%, *p* = 0.06) in a cohort of patients with elevated baseline severity of illness (Hines Severity Score Index ≥ 2) [10].

Our findings provide additional data regarding the real-world effectiveness of fidaxomicin that should encourage clinicians, hospitals, and third-party payors to utilize fidaxomicin as the preferred antimicrobial treatment option for CDI. The significant reduction in CDI recurrence observed in randomized controlled trials has now been replicated in a national sample utilizing real-world data. Participants in a randomized, controlled trial chose to go through an informed consent process, be randomized, and have scheduled follow-up visits or contacts. While necessary, these procedures can create a selection bias that creates study populations that may be different from people who chose not to participate in randomized, controlled trials. In addition, the data from severe CDI cases in the VA national cohort had cast some doubt on whether fidaxomicin decreased recurrence rates in a real-world setting. Our findings are from across multiple healthcare systems that are representative of the American population. Many investigators have evaluated the cost-effectiveness of fidaxomicin for CDI with the findings being mixed depending on the methods and data utilized [9,11,12,13,14,15,16,17,18,19]. A recent narrative review points out the need for data to shed light on the true attributable costs of CDI beyond what is captured in the existing literature [20]. The uncertainty regarding fidaxomicin’s real-world effectiveness created an environment where fidaxomicin use was restricted. A multicenter cohort of 244 patients from four Spanish hospitals observed that only 39% of fidaxomicin use was for a first episode [21]. The investigators did not evaluate any other CDI treatments, so we do not know how this compares to other treatment options. The study by Gallagher and colleagues also outlines a patient population where fidaxomicin use was limited by facility guidelines intended to guide “appropriate use” [9].

Our study has many limitations including being retrospective, which did not allow us to evaluate bowel movements or the necessity for additional CDI treatment at facilities that do not participate in Cerner Health Facts^®^. Using a retrospective design did allow us to evaluate the fidaxomicin and vancomycin impact on CDI recurrence relative to each other in several institutions outside of a clinical trial setting that may exclude populations that need CDI treatment and may bias results compared to a real-world setting. Our definition of recurrence was reliant upon ICD-9 or ICD-10 coding. However, past evaluations of the VA system have found that the coding for CDI is considered generally reliable [22,23]. Cerner HealthFacts^®^ also primarily has data from hospitals or hospital systems with limited outpatient data. This may have decreased the number of CDI recurrences documented, especially compared to the trials whose outpatient participants made up 40% of the study population [6,7]. However, rehospitalizations due to CDI recurrence may indeed represent a more clinically important outcome and better characterize fidaxomicin’s ability to prevent more severe outcomes in patients who have a CDI recurrence. Another potential reason for fewer recurrences in our cohort is that we did not allow for any prior CDI episodes within the previous year compared to others who included patients with previous episodes of CDI, one even within 90 days [4,5,6,7]. This also means that our findings cannot be extrapolated to patients who are being treated for a recurrent CDI episode.

## 4. Materials and Methods

### 4.1. Study Design

This retrospective study was conducted using the Cerner Health Facts^®^ Database (Cerner, Kansas City, MO, USA) to evaluate the impact of CDI treatment choice (fidaxomicin vs. vancomycin) on recurrence rates in hospitalized patients. The data in the Cerner Health Facts^®^ Database is extracted directly from the EMR (electronic medical record) from hospitals in which Cerner has a data use agreement. Encounters may include pharmacy, clinical and microbiology laboratory, admission, and billing information from affiliated patient care locations. All admissions, medication orders and dispensing, laboratory orders, and specimens are date and time-stamped, providing a temporal relationship between treatment patterns and clinical information. Cerner Corporation has established Health Insurance Portability and Accountability Act-compliant operating policies to establish de-identification for Health Facts. A total of 750 facilities contributed de-identified information on 69 million patients seen between January 2001 and July 2018.

We retrieved patients 18 years and older who were admitted between July 2011 through June 2018 and were diagnosed and treated for CDI with vancomycin or fidaxomicin. We used the International Classification of Diseases procedure codes, Ninth Revision (ICD-9) and Tenth Revision (ICD-10) to identify patients that were diagnosed with CDI (ICD-9 008.45, ICD-10 A04.7, A04.71, and A04.72) and treated with vancomycin or fidaxomicin. The first occurrence of CDI with treatment was designated as the index case. A severe CDI occurrence was determined by a white blood cell count of ≥15,000 cells/mL or serum creatinine greater than 1.5 mg/dL within 48 h of initiating CDI treatment with vancomycin or fidaxomicin as defined by the 2018 IDSA guidelines [24]. Fulminant cases were classified by identifying patients with hypotension, ileus, or megacolon as outlined in the 2018 IDSA CDI guidelines. We defined hypotension as a systolic blood pressure < 100 mmHg or diastolic blood pressure < 60 mmHg within 48 h of CDI treatment initiation. We used ICD-9 codes 560.1 or 560.3 and ICD-10 codes K56.0, K56.3, K56.7 to identify patients with ileus. We used ICD-9 code 564.7 as well as ICD-10 codes K59.31, K59.39 to identify patients with megacolon. Patients with CDI within 1 year prior to the index case were excluded. From the remaining index cases, a recurrence of CDI within 90 days of the index case was determined (Figure 2).

We also collected data on age, gender, race, census region, urban/rural facility status, white blood cell count, serum creatine level, systolic and diastolic blood pressure, presence of ileus or megacolon, antibiotic use prior to the index case, PPI and H2 antagonist use during the index case, and van Walraven scores for the Elixhauser Comorbidities prior to the index case [25,26]. We used inpatient and outpatient demographic, utilization, and comorbidity data from Cerner Health Facts^®^.

### 4.2. Statistical Analyses

Cohort analyses matched on propensity score for treatment with fidaxomicin were performed [27]. All patients were censored at 90 days after cohort entry to account for events that occur proximal to the incident exposure. Exposure was defined as having fidaxomicin at base cohort entry. To be categorized as fidaxomicin exposed, patients must not have fidaxomicin in the 1 year prior to base cohort entry and then had fidaxomicin at base cohort entry. Unexposed patients must not have a prescription for fidaxomicin in the 1 year prior to base cohort entry and then not prescribed fidaxomicin, rather prescribed vancomycin at base cohort entry. The intent-to-treat principle was utilized to categorize exposure during follow-up. Logistic regression models were used to create the propensity score for fidaxomicin exposure, which modeled the probability of fidaxomicin use at baseline [28]. These candidate covariates used were selected based on previous literature [29,30]. Nearest-number matching was performed with a caliper of 0.02. Patients treated with fidaxomicin were matched 1:1 to patients receiving vancomycin at base cohort entry. Standardized differences were calculated to assess the balance of covariates in the propensity score-matched groups. The standardized difference compares the difference in means in units of the pooled standard deviation [31]. Unlike tests of a statistical hypothesis, the standardized difference is not influenced by sample size. Standardized differences were calculated to evaluate any differences between groups. Hazard ratios and 95% confidence intervals (95% CI) were estimated using Cox proportional hazards model regressing survival on fidaxomicin exposure in the propensity-matched dataset. The proportional hazards assumption was examined by checking Schoenfeld residuals over time [32]. A plot that shows a non-random pattern against time is evidence of a violation of the proportional hazards assumption. To account for the matched nature of the dataset, robust variance estimators were calculated to account for clustering within matched sets [33]. Kaplan–Meier curves were constructed for the propensity-matched cohort. Statistical analyses were conducted using R, version 3.6.1 (R Foundation for Statistical Computing, Vienna, Austria).

## 5. Conclusions

Fidaxomicin was independently associated with decreased CDI recurrence compared to vancomycin, as defined by readmission for CDI within 90 days, for patients presenting with an index CDI episode using a national sample of multiple healthcare systems. This conclusion is limited by the lack of data from facilities outside of the Cerner Health Facts^®^ network and by the definition of CDI recurrence differing from those used in randomized, controlled trials. Since fidaxomicin is better than vancomycin for an index case of CDI, we recommend despite the limitations of this study that fidaxomicin might be a better treatment, but confirmatory studies are needed.

## Figures and Tables

**Figure 1 antibiotics-11-00295-f001:**
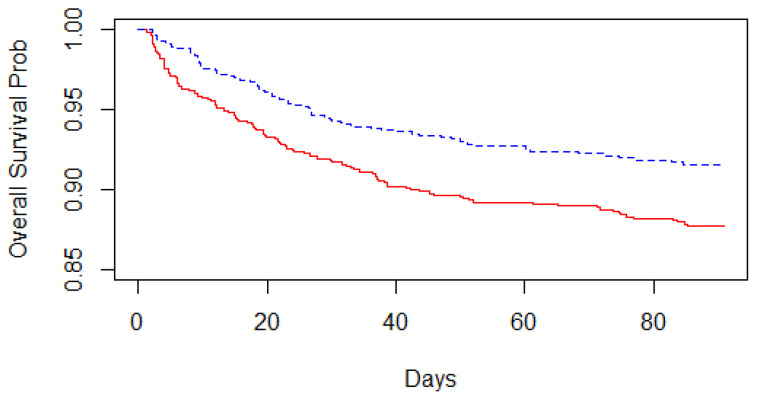
Adjusted Kaplan–Meier graph depicting the risk of CDI recurrence over 90 days. Vancomycin is represented by the solid red line and fidaxomicin by the dashed blue line. The results of the Cox Proportional Hazard model also revealed a statistically significantly decreased risk of CDI recurrence with fidaxomicin in the matched cohort (HR 0.67, 95% CI 0.50–0.90).

**Figure 2 antibiotics-11-00295-f002:**
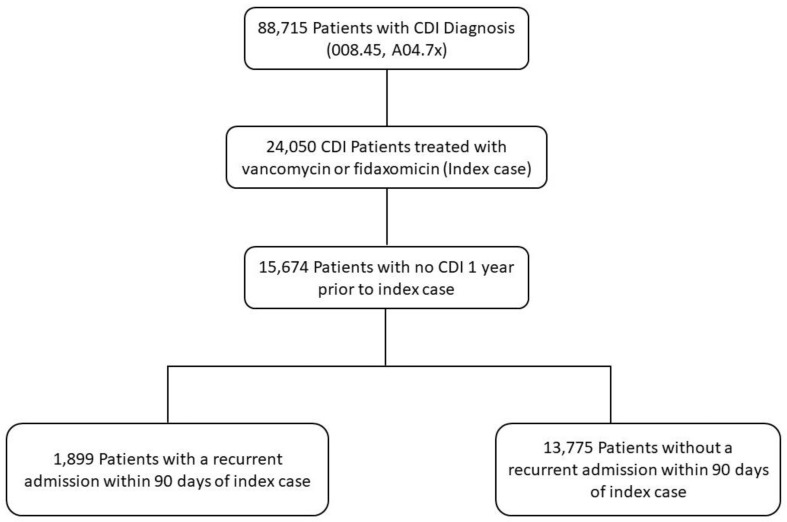
Data extraction diagram.

**Table 1 antibiotics-11-00295-t001:** Baseline characteristics (mean, standard devation unless otherwise noted).

	Entire Cohort			Matched Patients		
Variable	Vancomycin (*n* = 14,785)	Fidaxomicin (*n* = 889)	Standardized Difference	Vancomycin (*n* = 889)	Fidaxomicin (*n* = 889)	Standardized Difference
Age in years	66.88 (16.73)	64.17 (18.34)	−0.147	65.04 (17.87)	64.17 (18.34)	−0.047
van Walraven Score	16.68 (11.11)	15.57 (10.71)	−0.104	15.26 (10.47)	15.57 (10.71)	0.029
Male Gender (*n*, %)	6214 (42.0)	342 (38.5)	−0.073	331 (37.2)	342 (38.5)	0.025
Race (*n*, %)						
Caucasian	10,870 (73.5)	708 (79.6)	0.152	721 (81.1)	708 (79.6)	−0.036
African American	2802 (19.0)	115 (12.9)	−0.179	107 (12.0)	115 (12.9)	0.027
Other	1113 (7.5)	66 (7.4)	−0.004	61 (6.9)	66 (7.4)	0.022
Severe (*n*, %)	4152 (28.1)	246 (27.7)	−0.009	245 (27.6)	246 (27.7)	0.002
Fulminant (*n*, %)	89 (0.6)	1 (0.1)	−0.146	1 (0.1)	1 (0.1)	0.00
Prior Antibiotics (*n*, %)	7122 (48.2)	414 (46.6)	−0.032	394 (44.3)	414 (46.6)	0.045

**Table 2 antibiotics-11-00295-t002:** Characteristics by recurrence status (mean, standard devation unless otherwise noted).

	Entire Cohort			Matched Cohort		
Variable	No Recurrence (*n* = 13,775)	Recurrence (*n* = 1899)	*p*-Value	No Recurrence (*n* = 1594)	Recurrence (*n* = 184)	*p*-Value
Age in years	66.65 (16.85)	67.31 (16.75)	0.11	64.46 (18.15)	65.90 (17.72)	0.30
van Walraven Score	16.72 (11.16)	15.91 (10.50)	0.002	15.51 (10.57)	14.60 (10.70)	0.27
Male Gender (*n*, %)	5815 (42.2)	741 (39.0)	0.009	605 (38.0)	68 (37.0)	0.85
Race (*n*, %)			0.006			0.19
Caucasian	10,217 (74.2)	1361 (71.7)		1285 (80.6)	144 (78.3)	
African American	2505 (18.2)	412 (21.7)		192 (12.0)	30 (16.3)	
Other	1053 (7.6)	126 (6.6)		117 (7.3)	10 (5.4)	
Severe (*n*, %)	3765 (27.3)	633 (33.3)	<0.001	428 (26.9)	63 (34.2)	0.04
Fulminant (*n*, %)	82 (0.6)	8 (0.4)	0.44	2 (0.1)	0 (0.0)	1.00
Prior Antibiotics (*n*, %)	6460 (46.9)	1076 (56.7)	<0.001	702 (44.0)	106 (57.6)	<0.001

## Data Availability

Restrictions apply to the availability of these data. Data was obtained from Cerner Health Facts and are available from Cerner Health Facts if they approve the request.

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
