# Peer review of "The Risk of Clostridioides difficile Recurrence after Initial Treatment with Vancomycin or Fidaxomicin Utilizing Cerner Health Facts"

_antibiotics, 2022, doi:10.3390/antibiotics11030295_

Round 1

Reviewer 1 Report

The manuscript of R.G. Hall et al. titled "The risk of Clostridioides difficile recurrence after initial treatment with vancomycin or fidaxomicin utilizing Cerner Health Facts®" devoted to the problem of Clostridioides difficile infection (CDI) recurrences and effect of vancomycin or fidaxomicin, frequently used for treatment, on recurrence of CDI. The authors used data from the Cerner Health Facts database and statistical analysis. They concluded that fidaxomicin was independently associated with decreased CDI recurrence. Since the data are not particularly novel, the authors presented the study in the form of a brief report.

Although I did not find many major issues, I put here some minor issues and suggestions/typos aimed to improve the manuscript:

Line 2: Italisize Clostridioides difficile.
Line 4: I have doubts about the need to use ® in the title.
Abstract: I suggest stating the aim of the study more clear.
Line 15: I suggest putting the number of patients in the abstract.
Line 25: Omit [1]
Line 32: You started numeration from 2. Where the ref. 1?
Tables 1 and 2: What means numbers? Absolute numbers and percents? Should readers guess?
Tables 1 and 2: Add space between Wan and Walraven.
Line 81: 2.3 comes after 2.1. Did you miss 2.2? Where can I find section 2.2?
Figure 1: What means red and blue colors? Add it to the legend.
Line 183: Omit title of Fig. 2 "Data Extraction Diagram".
Line 193: What means "(1)"?

Author Response

Please see the attached Word file for our responses to your thoughtful comments.

Reviewer 2 Report

The review report is attached.

Author Response

(The authors gave the same response as above.)

Reviewer 3 Report

I liked the paper. I have some comments:

  • language/editing corrections lines: 6, 43, 46, 165, 177
  • please specify in the text (or ad citations) what do you mean by 'VanWalraven score'
  • how did you decide wheter it is non-severe, fulminant or severe CDI
  • please explain EMR

Author Response

(The authors gave the same response as above.)
